# Characteristics of Healthcare Workers Vaccinated against Influenza in the Era of COVID-19

**DOI:** 10.3390/vaccines9070695

**Published:** 2021-06-24

**Authors:** Giorgia Della Polla, Francesca Licata, Silvia Angelillo, Concetta Paola Pelullo, Aida Bianco, Italo Francesco Angelillo

**Affiliations:** 1Health Direction, Teaching Hospital, University of Campania “Luigi Vanvitelli”, Via Santa Maria di Costantinopoli 104, 80138 Naples, Italy; giorgia.dellapolla@unicampania.it; 2Department of Health Sciences, University of Catanzaro ‘‘Magna Græcia”, Viale Europa, 88100 Catanzaro, Italy; francesca.licata@studenti.unicz.it (F.L.); silvia.angelillo@studenti.unicz.it (S.A.); a.bianco@unicz.it (A.B.); 3Department of Experimental Medicine, University of Campania “Luigi Vanvitelli”, Via L. Armanni 5, 80138 Naples, Italy; concettapaola.pelullo@unicampania.it

**Keywords:** COVID-19 pandemic, healthcare workers, influenza vaccination, Italy, survey, vaccination coverage

## Abstract

Understanding the potential impact of COVID-19 on receiving influenza vaccination among healthcare workers (HCWs) is of utmost importance. The purposes of the present cross-sectional study were to describe the characteristics and to explore the predictors of receiving influenza vaccination among a large cohort of Italian HCWs in hospital settings. Information was collected through an anonymous questionnaire from December 2020 through January 2021. General and practice characteristics, perceived risk of seasonal influenza, attitudes towards efficacy and safety of influenza vaccination, and reasons behind the decision to be vaccinated against influenza were explored. Fewer than half (46.2%) of HCWs agreed that influenza is a serious illness and perceived the risk of getting infected with influenza, and concerns about the safety of the vaccination were significant positive predictors. Fewer than half of the respondents were not concerned at all about the efficacy (48.6%) and safety (49.8%) of influenza vaccination, and 51.9% reported that they have not received a seasonal influenza vaccine during the previous season. The most mentioned reason for receiving the influenza vaccine in the current season was that influenza and COVID-19 share some similar symptoms. Study results will aid policymakers in developing vaccination education programs, promotion of trust to address negative misconceptions, and to achieve future high coverage among this high-risk group.

## 1. Introduction

It is well-known that seasonal influenza can cause significant morbidity and mortality in most communities, with 3 to 5 million cases, and more than 290,000 to 650,000 respiratory deaths worldwide [1,2]. A substantial body of the literature has described that the differences in the risk of contracting an influenza infection may be related to varying levels of exposure, with healthcare workers (HCWs) being indicated as one of the main groups at risk [3]. Moreover, HCWs with influenza are an important source of infection for vulnerable patients, and in particular the hospitalized population has high rates of serious underlying illnesses, making influenza more dangerous in this setting [3].

The best public health strategy to prevent influenza is immunization through seasonal vaccines that are extremely safe, highly effective, and can reduce morbidity and mortality, especially if high coverage is achieved [4,5]. Despite this evidence and the efforts to encourage the vaccination, the body of data concerning the immunization of HCWs indicates that overall coverage continues to remain unacceptably low among this group [6,7,8]. In Italy, the Ministry of Health annually recommends that HCWs get vaccinated against influenza as an important protective action for them and to prevent transmission to their families, colleagues, and patients. The target goal is at least 75% coverage [9]. In the 2020–2021 season, several influenza vaccines were licensed in Italy: conventional trivalent vaccines, one adjuvanted trivalent vaccine, and quadrivalent vaccines. It is recommended to use the quadrivalent vaccine from 6 months of age up to 70 years and the adjuvanted trivalent vaccine in subjects >70 years. A dose of quadrivalent vaccine is recommended for HCWs.

Influenza vaccination was more important than ever during the 2020–2021 season since the novel pandemic coronavirus disease 2019 (COVID-19), caused by the severe acute respiratory syndrome coronavirus 2 (SARS-CoV-2), is still actively circulating worldwide. Moreover, the symptoms of both respiratory viral infections are similar, and they also share the same high-risk populations, including HCWs. Therefore, this is a significant issue. There is a paucity of these data, and understanding the potential impact of COVID-19 on receiving influenza vaccination among HCWs is of utmost importance. Considering this, the purposes of the present survey were to describe the characteristics and to explore the predictors of receiving influenza vaccination among a large cohort of Italian HCWs in hospital settings.

## 2. Materials and Methods

### 2.1. Study Design and Participants

A cross-sectional survey was conducted from 18 December 2020 to 18 January 2021 among HCWs. The target population was HCWs working in both clinical and non-clinical roles in five Teaching and non-Teaching Hospitals located in the cities of Catanzaro and Naples in the southern part of Italy.

### 2.2. Sampling Procedures

All HCWs who underwent influenza vaccination were approached and invited to join the survey while at the immunization center. To calculate a representative sample size of the target population, the assumption that 50% of respondents had not received a vaccine against seasonal influenza in the previous season was used, with a 95% confidence interval, and an allowable error of 5%. A minimum sample size of 385 HCWs was determined. This estimated sample size was adjusted for a non-response rate of 20%, yielding a final target sample population of 481 HCWs.

Information was collected through an anonymous questionnaire, and the respondents completed either a telephone interview by trained study personnel or a self-administered questionnaire. The questionnaire was pre-tested among a group of 20 HCWs, who were not part of the study sample, to evaluate whether the questions effectively captured the topic under investigation.

### 2.3. Study Tool

A four-section questionnaire was prepared. The first section concerned socio-demographic, occupational, and health-related characteristics, such as age, gender, marital status, type of occupation and practice, length of professional activity, and having underlying chronic medical conditions. The second section focused on attitudes, and the participants were asked to rate their agreement. The perceived risk of seasonal influenza according to three indicators (severity, risk of being infected as HCWs, and risk that HCWs pass the influenza virus on to their patients) was measured on a five-point Likert scale, ranging from 1 = strongly disagree to 5 = strongly agree; two statements about safety and efficacy of influenza vaccination were measured on a five-point Likert-type scale ranging from 1 = extremely to 5 = not at all; and one statement on the risk perception of getting infected with seasonal influenza was measured with a ten-point Likert-type scale ranging from 1 = low to 10 = high. The third section collected information on the reasons behind the participant’s decision to be vaccinated against influenza in the current and the previous season. In the last section, the main sources of information about influenza vaccination utilized, and the level of trust in those sources, was explored. HCWs were also asked whether they were interested in having further information about seasonal influenza vaccination.

### 2.4. Data Collection

Before administration of the survey, the participants were informed about the objectives of the study and were guaranteed that confidentiality and anonymity of the gathered information would be maintained and that no data identifying a responder were collected. They were also informed that participation was voluntary and that they can withdraw from the study whenever they choose without reprisal. All participants gave written, informed consent to participate. No compensation or other incentives was offered to the participants for their time spent.

### 2.5. Ethics

Institutional ethical approval was obtained for data collection by the Teaching Hospital of the University of Campania “Luigi Vanvitelli” Ethics Committee.

### 2.6. Statistical Analysis

Descriptive statistics were used for all data; continuous variables were described as means with standard deviations, whereas categorical variables were presented as frequencies. Initially, categorical and continuous variables were compared, respectively, by using the Chi-square test or Student t-test. Variables with a *p*-value < 0.25 in the univariate analysis were included in the final multivariate logistic regression models, and the significant level choices for the inclusion and elimination of the variables in the models were *p*-values of 0.2 and 0.4, respectively. Multivariable regression analysis assessed the relationship between the different characteristics and the three main following outcomes of interest: belief that influenza is a serious illness (Model 1); having received the influenza vaccine only in the current season (Model 2); having indicated that influenza and COVID-19 share similar symptoms as a reason to be vaccinated (Model 3). The following selected independent variables were included in all regression models: gender (female = 0; male = 1); age, in years (continuous); marital status (unmarried/separated/divorced/widowed = 0; married/cohabitant = 1); professional role (physician = 1; nurse = 2; other = 3); working area (medical = 1; surgical = 2; laboratory and diagnostic = 3; critical care/COVID-19 units = 4); length of practice, in years (continuous); having underlying chronic medical conditions (no = 0; yes = 1); perceived risk of getting infected with influenza (continuous); being concerned about efficacy of the influenza vaccination (yes = 0; no = 1); being concerned about safety of influenza vaccination (yes = 0; no = 1); belief that HCWs are at risk of getting infected with influenza (no = 0; yes = 1); belief that an infected HCW can pass the influenza virus on to their patients (no = 0; yes = 1); scientific journals, meetings and colleagues as sources of information about influenza vaccination (no = 0; yes = 1); and needing additional information regarding influenza vaccination (no = 0; yes = 1). Moreover, the variable having indicated that influenza and COVID-19 share similar symptoms as a reason to be vaccinated (no = 0; yes = 1) was included in Models 1 and 2; belief that influenza is a serious illness (no = 0; yes = 1) was included in the Models 2 and 3; having received the influenza vaccine only in the current season (no = 0; yes = 1) was included in Models 1 and 3; having been vaccinated regardless of COVID-19 (continuous) was included in Model 2. The analysis result of multivariate logistic regression is expressed as odds ratio (OR) with a 95% confidence interval (95% CI). All *p*-values were two-sided, and the values of 0.05 or less were considered to be statistically significant. All data were analyzed by STATA 15 software [10].

## 3. Results

Of the 843 HCWs approached, a total of 615 agreed to participate for a response rate of 72.9%. Table 1 showed the HCWs’ demographic and professional characteristics. More than half were females, the mean age was 45.3 years, more than half were physicians, the mean length of practice was 11.7 (1–42) years, almost half worked in the medical area, and one-fifth reported to have at least one chronic medical condition.

The participants showed a low level of perceived risk of getting infected with seasonal influenza, measured on a 10-point Likert-type scale, with a mean value of 4.3. Fewer than half (46.2%) of the HCWs believed that influenza is a serious illness; 79.8% and 75.4% believed that HCWs are at risk to get infected with influenza and that an infected HCW can pass the virus on to their patients, respectively. Regarding the vaccine, 48.6% and 49.8% of the respondents were not concerned at all about the efficacy and safety of influenza vaccination, respectively. Table 2 shows the results of the multivariate logistic regression analysis conducted to predict the characteristics associated with the different outcomes of interest. Perceived risk of getting infected with influenza and concerns about the safety of the influenza vaccination were significantly associated with the belief that influenza is a serious illness. Specifically, HCWs perceived to be at risk of getting infected with the disease had a 1.34 (95% CI 1.24–1.45) greater likelihood of believing that influenza is a serious illness compared with those who did not perceive themselves to be at risk. Respondents who believed that HCWs are at risk to get infected with influenza (OR = 3.75; 95% CI = 1.96–7.17) and those who were not concerned at all about the safety of the influenza vaccination (OR = 1.5; 95% CI = 1.03–2.18) were more likely to believe that influenza is a serious illness (Model 1 in Table 2).

More than half (51.9%) of the study participants reported that they had not received a seasonal influenza vaccine during the previous season. Among all respondents, the most mentioned reason for receiving the influenza vaccine in the current season was that influenza and COVID-19 share similar symptoms (31.6%), followed by the belief that the vaccine is useful to protect from seasonal influenza (25%), that the vaccine is safe (13.7%), and believing themselves to be at risk of contracting seasonal influenza (9.6%). Participants who did not believe that an infected HCW can pass the influenza virus on to their patients (OR = 0.47; 95% CI = 0.24–0.91), not having an underlying chronic medical condition (OR = 0.28; 95% CI = 0.14–0.56), having indicated that influenza and COVID-19 share similar symptoms as a reason to be vaccinated (OR = 3.93; 95% CI = 2.24–6.91), and those having not been vaccinated regardless of COVID-19 (OR = 0.04; 95% CI = 0.02–0.07) had a greater likelihood of having received the influenza vaccine only in the current season compared with those who had been vaccinated in both seasons (Model 2 in Table 2). Respondents of younger age (OR = 0.96; 95% CI = 0.94–0.98), those serving as physician (OR = 0.58; 95% CI = 0.35–0.95), those not believing that an infected HCW can pass the influenza virus on to their patients (OR = 0.16; 95% CI = 0.09–0.29), and those who declared to have been vaccinated for influenza both in the current and in the previous season (OR = 0.13; 95% CI = 0.08–0.21) were more likely to indicate the fact that influenza and COVID-19 share similar symptoms as a reason to be vaccinated. Respondents who stated that they had received information on influenza vaccination from scientific journals, meetings, and colleagues were almost two times more likely to be vaccinated for the above-mentioned reason (OR = 1.83; 95% CI = 1.11–3.04) (Model 3 in Table 2).

Almost all the responding HCWs (93.7%) indicated that they had received information about the vaccination against seasonal influenza. The main sources of information mentioned by the HCWs were scientific journals (52.4%), followed by colleagues (34.3%), the Internet (30.9%), news broadcasts and mass media (28.3%), and meetings/conferences (11.5%). Fewer than one-third of the respondents (28.6%) wished to receive additional information.

## 4. Discussion

The study results showed that the attitudes, especially influenza’s perceived severity that was relatively lower than the perceived risk for infection susceptibility and to pass the influenza virus on to patients, deserve particular attention. Moreover, widespread negative vaccine attitudes have been observed since more than half of the HCWs were concerned or uncertain regarding the efficacy and safety of the influenza vaccine, although such characteristics had a non-statistically significant impact on practicing vaccination behaviors. This observation is particularly worrisome and echoes some previous studies showing that the trust in the safety of the influenza vaccine is considered one of the main factors influencing vaccination uptake [11,12,13]. It is imperative that public health strategies explicitly promote communication and educational campaigns for HCWs to correct misinformation regarding influenza vaccines that influence the uptake among HCWs.

A sizeable proportion of HCWs that had not been vaccinated for seasonal influenza in the previous season (51.9%) shifted their behavior to receive the vaccination in this season. It should be noted that, unsurprisingly, the COVID-19 pandemic has had an impact on influenza vaccine acceptance among this group of HCWs. Indeed, the finding of a higher level of self-reported influenza vaccine coverage in the current season is explained by the results of the multivariate logistic regression analysis. Indeed, it has been observed that the perceived risk for the respondents to pass the influenza virus on to patients at their healthcare facility and having indicated that influenza and COVID-19 share similar symptoms as a reason to be vaccinated were the most significant predictors for having received the influenza vaccine in the current season among the HCWs who had not received the vaccination in the latest season. This finding is consistent with the results of several previous studies, although most used very different methodologies and populations, showing an increase in those who were very likely to accept vaccination for influenza [14,15,16,17] and of actual behavior [12,18,19].

Several findings on the associations between the HCWs’ socio-demographic, professional, and anamnestic characteristics and the different outcomes of interest provided valuable insights. Specifically, respondents of younger age and those serving as physicians indicated that the fact that the symptoms of seasonal influenza can be very similar to those of COVID-19 made them more likely to receive the influenza vaccine. This year, for the first time, the government in Italy has provided a free annual seasonal influenza vaccination for people from 60 years of age; therefore, those younger were less likely to be routinely vaccinated. The association between vaccine coverage and age is almost constant among the population in different geographic areas [14,20,21]. Moreover, those who did not have a chronic condition had a greater likelihood of having received the vaccine only in the present season compared to HCWs who had been vaccinated in both seasons. Not having chronic conditions was a predictor of vaccine uptake in this season, and those HCWs who did not have a condition were afraid of COVID-19 and had received the influenza vaccine because influenza and COVID-19 share similar symptoms. Possible explanations for this are that those who have concomitant chronic conditions are one of the more vulnerable groups making themselves more conscious of the necessity to be vaccinated annually and, therefore, more likely to accept the free annual influenza vaccination. Previous investigations have found higher vaccination coverage in patients with chronic conditions [22,23,24]. Finally, professional role disparity in the coverage rate may also be explained by differing professional recommendations and perceptions. This result agreed with those of several previously conducted studies, which showed the marked difference regarding the coverage for seasonal influenza vaccination with physicians being more likely to adhere to recommendations than other healthcare professionals [25,26,27,28]. This latter observation underlined the urgent need for nurses’ continuing education beginning at the undergraduate level in the field of vaccination.

In the multivariate logistic regression analysis, it has been observed that an additional variety of variables significantly influenced the different outcomes of interest. In particular, scientific journals, meetings, and colleagues were the leading sources of information on vaccination, and it was revealed to be a significant predictor of an outcome of interest. Indeed, the results revealed that HCWs who had used these sources were more likely to be vaccinated since the symptoms of seasonal influenza can be very similar to those of COVID-19. It should be stressed that these findings highlight the importance of scientific sources. The trust of the sampled HCWs in this source of information, in particular scientific journals, has a strategic value in acquiring adequate and correct information and suggests that they must use these sources as a clue to adhere to vaccination recommendations to improve uptake of influenza vaccine in this high group at risk. The current findings should also be considered alongside existing scientific evidence from previous studies among HCWs showing that the use of these sources of information is known to be consistently and significantly associated with a higher level of vaccination knowledge, more positive attitudes, perceptions of the need to receive vaccines, and higher adherence to vaccination recommendations by governmental and health organizations [29,30,31,32,33,34]. The fact that about one-third of the respondents searched the Internet and media for information about vaccination is of concern since it is not easy to filter irrelevant or wrong information from these sources. Therefore, HCWs need to be protected from misinformation and rely only on scientific findings that are approved by health experts. This is consistent with previous studies since using media and the Internet as primary information sources has been observed to be associated with an increased likelihood for hesitancy towards vaccination among different groups of individuals compared with those influenced by a medical authority [35,36,37,38,39,40]. Finally, almost one-third of the participants reported that they would be seeking additional information on influenza vaccination.

This study should be carefully interpreted within the context of its methodological limitations. First, the study was based on a cross-sectional design; hence, only associations can be presented, and causality cannot be concluded from the findings because temporal sequence cannot be established. Second, this study was conducted in two geographic areas; hence, generalizations of the results presented herein should be made cautiously to the general population of hospital-based HCWs in Italy. Third, the vaccination data were collected using a self-reporting questionnaire, and this may have allowed HCWs to respond inaccurately. For example, the subjects’ influenza immunization behavior in the previous year was not confirmed based on their medical records and, therefore, there may be a potential overestimation of the compliance to immunization due to social desirability.

In conclusion, in the era of COVID-19, more than half of the unvaccinated HCWs in the previous year had received the influenza vaccination in the current season, and the main reasons were related to the pandemic. These results and insights are essential, as they will aid policymakers in developing vaccination education programs and promotion of trust by the health authorities to address negative misconceptions about seasonal influenza vaccine and to achieve future high coverage among this high-risk group.

## Figures and Tables

**Table 1 vaccines-09-00695-t001:** The main characteristics of the study population.

Characteristic	Total	
	N	%
Age, years	45.3 ± 12.9 (22–70) *	
Gender		
Female	362	59
Male	252	41
Marital status		
Married/cohabitant	397	64.5
Unmarried/widowed/separated/divorced	218	35.5
Professional role		
Physician	327	53.4
Nurse	193	31.5
Other	92	15.1
Current working area		
Medical	279	46.9
Surgical	124	20.8
Laboratory and Diagnostic	99	16.6
Critical care/COVID-19 units	94	15.8
Length of practice in years	11.7 ± 10.8 (1–42) *	
Having underlying chronic medical conditions		
No	486	79
Yes	129	21
Having been vaccinated against influenza in the previous year		
No	318	51.9
Yes	295	48.1

* Mean ± Standard Deviation (Range). The number for each item may not add up to the total number of the study population due to missing values.

**Table 2 vaccines-09-00695-t002:** Multiple logistic regression analysis according to several explanatory variables.

Variable	OR	SE	95% CI	*p*
Model 1. Belief that influenza is a serious illness (Sample size = 586)				
Log likelihood = −329.38, χ^2^ = 149.99 (8 df), *p* < 0.0001				
Perceived risk of getting infected with influenza	1.34	0.05	1.24–1.45	<0.001
Belief that HCWs are at risk of getting infected with influenza	3.75	1.24	1.96–7.17	<0.001
Not being concerned at all about the safety of the influenza vaccination	1.5	0.29	1.03–2.18	0.036
Having received the influenza vaccine only in the current season	1.47	0.30	0.99–2.18	0.055
Belief that an infected HCW can pass the influenza virus on to their patients	1.60	0.45	0.92–2.77	0.094
Scientific journals, meetings, and colleagues as sources of information	1.38	0.31	0.89–2.14	0.148
Needing additional information about influenza vaccination	1.33	0.29	0.87–2.05	0.181
Length of practice in years	1.01	0.01	0.99–1.02	0.358
Model 2. Having received the influenza vaccine only in the current season (Sample size = 549)				
Log likelihood = −206.98, χ^2^ = 345.36 (8 df), *p* < 0.0001				
Not having underlying chronic medical conditions	0.28	0.1	0.14–0.56	<0.001
Having indicated that influenza and COVID-19 share similar symptoms as a reason to be vaccinated	3.93	1.13	2.24–6.9	<0.001
Not having been vaccinated regardless of COVID-19	0.04	0.01	0.02–0.07	<0.001
Belief that an infected HCW cannot pass the influenza virus on to their patients	0.47	0.16	0.24–0.91	0.026
Belief that influenza is not a serious illness	0.65	0.17	0.39–2.11	0.114
Older	1.02	0.02	0.99–1.06	0.118
Length of practice in years	0.97	0.02	0.94–1.01	0.154
No perceived risk of getting infected with influenza	0.95	0.05	0.86–1.05	0.35
Model 3. Having indicated that influenza and COVID-19 share some similar symptoms as a reason to be vaccinated (Sample size = 593)				
Log likelihood = −279.71, χ^2^ = 220.57 (9 df), *p* < 0.0001				
Having received the influenza vaccine both in the current and in the previous season	0.13	0.03	0.08–0.2	<0.001
Belief that an infected HCW cannot pass the influenza virus on to their patients	0.16	0.04	0.09–0.29	<0.001
Younger	0.96	0.01	0.94–0.98	0.001
Scientific journals, meetings, and colleagues as sources of information	1.83	0.47	1.11–3.04	0.018
Professional role				
Physician	1.00 *			
Nurse	0.58	0.14	0.35–0.95	0.032
Other	0.68	0.21	0.37–1.25	0.218
Belief that HCWs are at risk of getting infected with influenza	1.83	0.6	0.96–3.5	0.065
Being married or cohabitant	1.61	0.44	0.94–2.76	0.085
Having underlying chronic medical conditions	1.52	0.44	0.86–2.68	0.149

* Reference category.

## Data Availability

The data presented in this study are available on request from the corresponding author.

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
