# Peer review of "Characteristics of Healthcare Workers Vaccinated against Influenza in the Era of COVID-19"

_vaccines, 2021, doi:10.3390/vaccines9070695_

Round 1

Reviewer 1 Report

In Italy, which influenza immunizations are available?  Did all participants get the same immunization?  Are they offered immunizations such as the Live-Attenuated Influenza Vaccine (inhaled)?  I think that the authors should include this in the presentation of the material so that readers can determine if that is similar to the options that are offered in their settings.  

Sampling Procedures:

Line 68 -change suggested below

All HCWs who underwent influenza vaccination were approached and invited to join the survey while at the immunization center. 

Line 73 - 

The estimated sample size was adjusted for a non-73 response rate of 20%, yielding a final target sample population of 481 HCWs.  - a 20% non-response rate assumes a much higher rate of return on samples than is usually achieved.  Ultimately, you only received a response rate of 72.9%.

Discussion - this sentence is not well written - I think you want to say" Public health strategies should explicitly look to communicate and educate health care workers to correct misinformation regarding influenza vaccines that influence the uptake among HCWs.

Above all, it is imperative that public health officials’ strategies promote explicitly in communication and educational campaigns  around the influenza vaccine for correcting misinformation and improving the uptake to  maximize the compliance with government guidelines. 

Line 245 remove the word "the" prior to COVID-19 as it makes the sentence grammatically incorrect.

Possible explanations for this are that those suffering concomitant certain chronic conditions they are one of the more vulnerable group making themselves more conscious of the necessity to be vaccinated annually and, therefore, more likely to accept the free annual influenza vaccination.   Change to "

Possible explanations for this are that those suffering concomitant chronic conditions are one of the more vulnerable groups making themselves more conscious of the necessity to be vaccinated annually and, therefore, more likely to accept the free annual influenza vaccination. 

I am very interested on the generalizability of the results to future years.  Since Italy made major changes to the funding and availability of the influenza vaccine during this season, will this change (free availability) continue in the future.

Line 254 change "who" to "that"

Line 257 nurses' continuing education training beginning in at the undergraduate level in the field of vaccination.

I would be very interested to see if these findings are durable as COVID-19 wanes especially since the reason that additional uptake in immunization occurred due to the pandemic, in some cases.

Author Response

  1. As suggested, in the Introduction we have specified the influenza vaccines licensed in Italy in the 2020-2021 season and the recommendation for HCWs.
  2. (first version Line 68) As suggested, we have made the required change.
  3. (first version Line 73) In response to the point regarding the response rate, the achieved response rate was lower (9%) that the expected (80%), but the number of participants was considerably higher (615) that the estimated number (481).
  4. As suggested, in the Discussion section we have made all changes.

Reviewer 2 Report

This is an interesting and timely manuscript reporting on a cross-sectional analysis of the predictors of receiving the influenza vaccine during the Covid-19 pandemic among Italian health care workers. This topic could generate hypothesis for future investigations on ways to increase the decision of this population to receive the influenza vaccine.   The manuscript has a clear research justification with a logical rationale in the introduction. The manuscript data seem novel and build upon other cited research. The data collection and analysis approach seem reasonable to address the research aims.  The paper is generally well-written but I have some questions and comments.  

Major comments:

  • Approximately line 23. Adding some specific data here would help the reader.
  • Line 71. From where did this 50% come?
  • First paragraph of the Discussion section should start with most important and novel findings rather than a restatement of information from the introduction.  Start with paragraph 2:  "Study results show that..."  Moreover, the last sentence of the current first discussion paragraph does not make sense to me.
  • Line 212.  "...such characteristics had non-significant impact..." Be careful not to assume cause and effect. All you have is a statistical relationship based on a cross-sectional analysis. Moreover, if it is non-significant then by your own statistical approach this would not even be a relationship, let alone an impact.
  • Lines 27-28 and 215-218.  Was this assessed in this investigation? If not, this should not be a conclusion of this study.

Minor comments:

  • In the abstract, line 21 and elsewhere.  "Less" should be replaced with "fewer" to describe a lower number of countable nouns. Please check the entire manuscript as this is a minor, yet meaningful change to be made.
  • Line 55, "...of these data..."
  • Table 1. "N" is near the top of the column but not all of those data below are sample sizes.
  • Line 158 "Table 2 shows..."
  • Line 177.  "Participants who..."
  • Line 207.  "...especially the disease’s..." Please specify that you are referring to influenza.
  • Line 247. "suffering" is subjective. Replace with "who have"
  • Lines 276-278. I found this part of that sentence to be quite confusing.

Author Response

Major comments

  1. (first version Line 23) As suggested, in the Abstract we have added more specific results.
  2. (first version Line 71) In response to the point regarding the “50%”, we have clarified that it is referred to respondents who had not received a vaccine against seasonal influenza in the previous season.
  3. As suggested, in the Discussion section we have deleted the first paragraph.
  4. (first version Line 212) It is well-known that one of the main limitations of the cross-sectional studies, as we have indicated in the paragraph of the methodological limitations, is that only associations can be presented, and causality cannot be concluded from the findings because temporal sequence cannot be established. Therefore, we were not able to assume cause and effect and we have added the word “non-statistical” in the sentence.
  5. (first version Lines 27-28 and 215-218) As suggested, the sentence has been modified.

Minor comments

As suggested, all minor corrections have been made.

Reviewer 3 Report

Estimated Authors,

I've read with great interest the present Italian multi-centric survey on the characteristics of healthcare workers vaccinated against influenza in the era of COVID-19. The study is properly designed, the methods are clearly reported as well as the main results. Discussion is properly written and includes up-to-date references.

In summary, I endorse the acceptance of this paper as it is, without any further amendment / improvement.

Author Response

Thank you for the extremely positive tone of your comments.

Round 2

Reviewer 2 Report

You made good changes to your manuscript.